# Bacterial TonB-dependent transducers interact with the anti-σ factor in absence of the inducing signal protecting it from proteolysis

Sarah Wettstadt[1‡], Francisco J. Marcos-Torres[1‡], Joaquín R. Otero-Asman[1], Alicia García-Puente[1], Álvaro Ortega[2], María A. Llamas [1] *

1 Department of Biotechnology and Environmental Protection, Estación Experimental del Zaidín-Consejo Superior de Investigaciones Científicas, Granada, Spain, 2 Department of Biochemistry and Molecular Biology B and Immunology, Faculty of Chemistry, University of Murcia, Regional Campus of International Excellence Campus Mare Nostrum, Murcia, Spain

‡ These authors contributed equally to this work and share first authorship.
* marian.llamas@eez.csic.es

**Data Availability Statement:** All relevant data are within the paper and its Supporting Information

## Abstract

Competitive bacteria like the human pathogen *Pseudomonas aeruginosa* can acquire iron from different iron carriers, which are usually internalized via outer membrane TonB-dependent receptors (TBDRs). Production of TBDRs is promoted by the presence of the substrate. This regulation often entails a signal transfer pathway known as cell-surface signaling (CSS) that involves the TBDR itself that also functions as transducer (and is thus referred to as TBDT), a cytoplasmic membrane-bound anti-σ factor, and an extracytoplasmic function σ ($\sigma^{ECF}$) factor. TBDTs contain an extra N-terminal domain known as signaling domain (SD) required for the signal transfer activity of these receptors. In the current CSS model, presence of the signal allows the interaction between the TBDT and the anti-σ factor in the periplasm, promoting the proteolysis of the anti-σ factor and in turn the $\sigma^{ECF}$-dependent transcription of response genes, including the TBDT gene. However, recent evidence shows that $\sigma^{ECF}$ activity does not depend on this interaction, suggesting that the contact between these 2 proteins fulfills a different role. Using the *P. aeruginosa* Fox CSS system as model, we show here that the SD of the FoxA TBDT already interacts with the C-terminal domain of the FoxR anti-σ factor in absence of the signal. This interaction protects FoxR from proteolysis in turn preventing transcription of $\sigma^{FoxI}$-dependent genes. By structural modeling of the FoxR/FoxA$^{SD}$ interaction, we have identified the interaction sites between these 2 proteins and provide the molecular details of this interaction. We furthermore show that to exert this protective role, FoxA undergoes proteolytic cleavage, denoting a change in the paradigm of the current CSS model.

files. Raw data is available at the Mendeley Data repository (Mendeley Data, V1, doi:10.17632/nxh4c8ymnn.2).

**Funding:** This work was funded by MCIN/AEI/10.13039/501100011033 Spanish agency with projects BIO2017-83763-P and PID2020-115682GB-I00 to MLL. JOA was supported by the Spanish Ministry of Economy through an FPI fellowship (BES-2013-066301) and AO with grant PID2021-122202OB-I00. The funders had no role in study design, data collection and analysis, decision to publish, or preparation of the manuscript.

**Competing interests:** The authors have declared that no competing interests exist

**Abbreviations:** ASD, anti-σ domain; AUC, analytical ultracentrifugation; CSS, cell-surface signaling; CTD, C-terminal domain; FPE, filamentous phage extrusion; PAE, predicted alignment error; PMF, proton motive force; RIP, regulated intramembrane proteolysis; RMSD, root mean square deviation; SD, signaling domain; STN, secretin and TonB-dependent receptors N-domain; S1P, site-1 protease; S2P, site-2 protease; TBDR, TonB-dependent receptor; TBDT, TonB-dependent transducer; T2SS, type 2 secretion system; T3SS, type 3 secretion system; T4P, type 4 pili system.

## Introduction

Iron is essential for life. However, despite being one of the most abundant elements on Earth, the bioavailability of iron is very limited because in our oxygen-rich atmosphere this metal readily oxidizes and precipitates in insoluble hydroxides. To acquire iron, bacteria produce and secrete high-affinity iron-scavenging compounds known as siderophores that keep iron in a chelated soluble state. Gram-negative bacteria take up iron-siderophore complexes via specific outer membrane receptors of the TonB-dependent receptor family (TBDR). TBDRs belong to the outer membrane β-barrel protein superfamily and share a common fold: a C-terminal 22-stranded antiparallel transmembrane β-barrel domain that forms a pore, and an N-terminal globular domain named plug, hatch, or cork (S1A Fig) [1]. The plug domain contains a mixed four-stranded β-sheet that is located inside the C-terminal barrel domain, occluding the pore in absence of the siderophore. Transport of the iron-siderophore complex requires energy in the form of proton motive force (PMF) and the TonB-ExbBD cytoplasmic membrane protein complex to transduce the energy to the outer membrane (S1B Fig) [1]. TonB is the protein of the complex that interacts with TBDRs in a 6-amino acids region located before the plug region termed the TonB box (S1A Fig).

Besides producing their own siderophores, most competitive bacteria use piracy strategies to acquire essential components like iron, thus increasing the capacity of the bacteria to thrive in many different environments. This is the case of the human pathogen *Pseudomonas aeruginosa*, which is very efficient in acquiring siderophores produced by other microorganisms (denoted xenosiderophores) and iron carriers from the host [2]. This capacity resides in the production of 34 different TBDRs, many of which are involved in the transport of iron-containing compounds [3,4]. Importantly, *P. aeruginosa* adjusts the production of TBDRs to the presence of their substrate. This regulation often entails a signal transduction circuit named cell-surface signaling (CSS) that involves the TBDR itself, a cytoplasmic membrane-bound anti-σ factor, and an extracytoplasmic function σ (σ^ECF) factor in the cytosol (S1B Fig) [5–7]. Binding of the substrate to the receptor is transduced to the anti-σ factor in the periplasm resulting in the liberation of the σ^ECF factor in the cytosol. The σ^ECF factor then binds to the RNA polymerase (RNAP) and promotes the transcription of signal-response genes, including that of the receptor gene (S1B Fig), thus increasing the capacity of the bacteria to transport the substrate. TBDRs with this dual function in transport and signaling have been named TonB-dependent transducers (TBDTs) [8]. TBDTs contain an extra N-terminal domain known as signaling domain (SD) that is not present in TBDRs not involved in signaling. This domain is connected to the plug domain by a periplasmic flexible loop that includes the TonB box region (S1A Fig) [9,10]. The SD consists of 2 α-helices positioned side by side which are sandwiched between 2 β-sheets with the sheets made of 2 and 3 β-strands (S1A Fig) [9,11,12]. The structure of this domain resembles the periplasmic domain of outer membrane secretins of bacterial secretion systems, a domain that has been denoted STN (from secretin and TonB-dependent receptors N-domain; SMART accession number SM00965). STN domains interact with each other, with STN domains from other secretins, and with the periplasmic domains of cytoplasmic membrane proteins [13,14]. Similarly, the SD of a TBDT interacts with the periplasmic C-terminal domain of its cognate CSS anti-σ factor.

CSS anti-σ factors are cytoplasmic membrane-anchored proteins that contain 3 domains: a cytosolic N-terminal domain that binds the σ^ECF factor blocking its interaction with the RNAP in absence of the CSS signal and is thus known as anti-σ domain (ASD), a single-pass transmembrane helix, and a periplasmic C-terminal domain (CTD) that receives the signal from the TBDT [5–7]. The crystal structure of the ASD of the PupR CSS anti-σ factor showed that it folds into a three-helix bundle and forms a dimer in solution [15]. PupR seems to lack an

ordered fourth helix found in cytosolic anti-σ factors like ChrR or RslA. This helix serves to block the interaction of the $\sigma^{ECF}$ factor with the RNAP. Therefore, it has been proposed that CSS anti-σ factors prevent the $\sigma^{ECF}$ factor/RNAP interaction by sandwiching the $\sigma^{ECF}$ factor between the ASD dimer thus tethering the $\sigma^{ECF}$ factor to the membrane and limiting access for the RNAP [15]. The structure of the CTD of PupR has also been solved and revealed that this domain can be divided into 2 subdomains; an N-terminal subdomain close to the cytoplasmic membrane, and a C-terminal subdomain that shows high structural similarity to the STN domain of secretins and the SD of TBDTs [16].

In the current CSS model, signal perception promotes the interaction between the SD and the anti-σ factor in the periplasm. This model is mainly based on earlier results obtained with the *Escherichia coli* Fec system. By introducing mutations that compromise the interaction between the TBDT FecA and the anti-σ factor FecR, $\sigma^{FecI}$ activity was reduced in response to the inducing signal ferric citrate [17,18]. However, our more recent results have shown that the SD/anti-σ factor interaction can be abolished without inhibiting $\sigma^{ECF}$ activity [19], which suggests that TBDTs could function in a different way than initially thought. Importantly, signal perception does not induce alterations in the overall structure of the SD and only modifies the position of this domain [10,12,20]. The region that connects the SD to the plug and β-barrel is long and flexible and has been proposed to enable movement of the SD in the periplasm upon signal perception, thereby promoting the interaction with the anti-σ factor. However, although the position of the SD with respect to the β-barrel in the outer membrane indeed changes in response to the inducing signal, the SD does not extend further into the periplasm [10,20]. This raises the question of how the signal is transmitted from the outer membrane to the cytosol to modify the activity of the CSS $\sigma^{ECF}$ factor. Our previous work using the *P. aeruginosa* Fox CSS system as a model has shed light on this matter by showing that signal transmission entails the regulated proteolysis of the anti-σ factor FoxR (S1B Fig) [21–23]. The TBDT of this system, the FoxA protein, recognizes and imports the xenosiderophore ferrioxamine B, among other structurally related compounds [24–27]. Besides being processed in presence of the signal, FoxR undergoes self-cleavage prior to signal recognition by an NO-acyl shift between glycine-191 and threonine-192 (GT site) [22,28]. This process separates the protein into 2 domains (FoxR$^N$ and FoxR$^C$, respectively) that are both required for proper signal transduction in response to ferrioxamine [22]. In our model (S1B Fig), presence of ferrioxamine B leads to the degradation of FoxR$^C$ in a process that involves the proteolytic activity of the C-terminal processing periplasmic proteases Prc and CtpA [23]. This allows the regulated intramembrane proteolysis (RIP) of the FoxR$^N$ domain by a still unknown site-1 protease (S1P) and the RseP site-2 protease (S2P) [21–23]. It is at present unclear how the proteolysis of FoxR is initiated in response to ferrioxamine B. One possibility is that the proteases are activated in response to the signal. However, CSS anti-σ factors lacking part of the C-terminal domain can be processed in absence of the inducing signal [19], which indicates that the proteases are active and functional in non-inducing conditions. This is in accordance with Prc, CtpA, and RseP being required for several other processes in the cell not related to iron acquisition or anti-σ factor proteolysis. Therefore, we hypothesize that FoxR is protected from proteolysis in the absence of ferrioxamine while presence of the siderophore exposes the protease-sensitive domains of the protein, and that the signaling domain of FoxA (FoxA$^{SD}$) plays a role in this process. To get more insights into the CSS mechanism, we have analyzed in this work the interaction between the SD of the *P. aeruginosa* FoxA TBDT and the CTD of the FoxR anti-σ factor and its effect on $\sigma^{FoxI}$-mediated transcription both in vivo and in vitro. We show that overproduction of the SD of several *Pseudomonas* TBDTs blocks signaling and proteolysis of their cognate anti-σ factor, and that co-expression of FoxR with FoxA$^{SD}$ stabilizes the anti-σ factor. Moreover, we have determined that FoxR$^C$ is the domain that interacts with FoxA$^{SD}$

and we provide the molecular details of this interaction. By structural modeling of the FoxR/FoxA$^{SD}$ interaction, we have identified the interaction sites between these 2 domains. Finally, we show that FoxA$^{SD}$ is cleaved in vivo likely by the periplasmic protease CtpA. Based on these results, we propose a new model for CSS in which the SD of the TBDT and the anti-σ factor interact in the absence of the inducing signal that prevents the proteolysis of the anti-σ factor and thus blocks the activation of the CSS pathway.

## Results

### Overproducing the signaling domain (SD) of TBDTs inhibits CSS activity

To assess the effect of the SD of the TBDT on CSS activity, the SD of several *P. aeruginosa* (Pa) and *Pseudomonas putida* (Pp) TBDTs were overproduced from plasmid. All constructs used contain the signal sequence of the corresponding protein (see S1 Table) to allow export of the SD into the periplasm. CSS activity was analyzed using CSS-dependent transcriptional fusions in which the promoter region of the TBDT gene, which is one of the genes transcribed by the σ$^{ECF}$ factor in response to the CSS signal (S1B Fig) [5,6], was placed in front of the *lacZ* gene. By measuring β-galactosidase activity, we followed the activation of the CSS signaling pathway, which involves the release of the σ$^{ECF}$ factor and transcription from the TBDT gene promoter. We analyzed the activities of the Fox, Fiu, and Iut CSS systems, which respond to the xenosi-derophores ferrioxamine, ferrichrome, and aerobactin, respectively, and that of the *P. aeruginosa* Hxu system that responds to the presence of haem in the extracellular medium [21,29]. Overproducing each of the SDs completely inhibited CSS activities and resulted in bacteria unable to respond to the corresponding CSS-inducing signal (Fig 1A).

Then, we performed western blot analysis of the anti-σ factor component of the *P. aeruginosa* Fox and the *P. putida* Iut CSS systems (the FoxR and the IutY proteins, respectively) using N-terminally HA-tagged proteins produced from plasmid. These analyses showed that overproducing the SD of the cognate TBDT (FoxA and IutA, respectively) inhibits the RIP of these proteins in response to the inducing signal (Fig 1B). This is in agreement with the bacteria being unable to respond to the siderophore (Fig 1A) because blocking the RIP mechanism is known to prevent the liberation and activation of the σ$^{ECF}$ factor (S1B Fig) [6,21]. Together, these results indicate that the SD of the TBDT can protect the anti-σ factor from RIP.

### The signaling domain of FoxA stabilizes FoxR in vivo

To further understand the function of the SD of the receptor in the CSS pathway, we analyzed the stability of the *P. aeruginosa* FoxR anti-σ factor in strains overproducing FoxA$^{SD}$ by western blot using a polyclonal anti-FoxR antibody (S2 Fig). Because the antibody was not able to detect the chromosomally produced FoxR protein (S3A Fig), we aimed to detect FoxR proteins expressed constitutively from a P*tac* promoter on the pMMB67EH plasmid. Three different FoxR proteins were co-expressed with FoxA$^{SD}$ in a *P. aeruginosa* Δ*foxR* mutant: the wild-type protein, and N- and C-terminal HA-tagged proteins, which we have used in previous studies to detect FoxR using an anti-HA antibody [22,28] (S2B and S2C Fig, FoxR, HA-FoxR, and FoxR-HA, respectively). Importantly, the 3 proteins were able to complement the phenotype of a Δ*foxR* mutant (S3B Fig), indicating that they are functional. Strains were grown in iron-restricted medium supplemented with ferrioxamine and proteins were immunoblotted using polyclonal antibodies raised against the periplasmic domain of FoxR (S2B Fig, upper panel) or the SD of FoxA (S2B Fig, middle panel). Samples were also immunoblotted against the OprL protein as a loading control (S2B Fig, lower panel). In the strain expressing the non-tagged FoxR protein, protein bands were not detectable when FoxA$^{SD}$ was absent (S2B Fig, lane 2). When the 2 proteins were co-expressed, 3 FoxR protein bands were detected (S2B Fig,

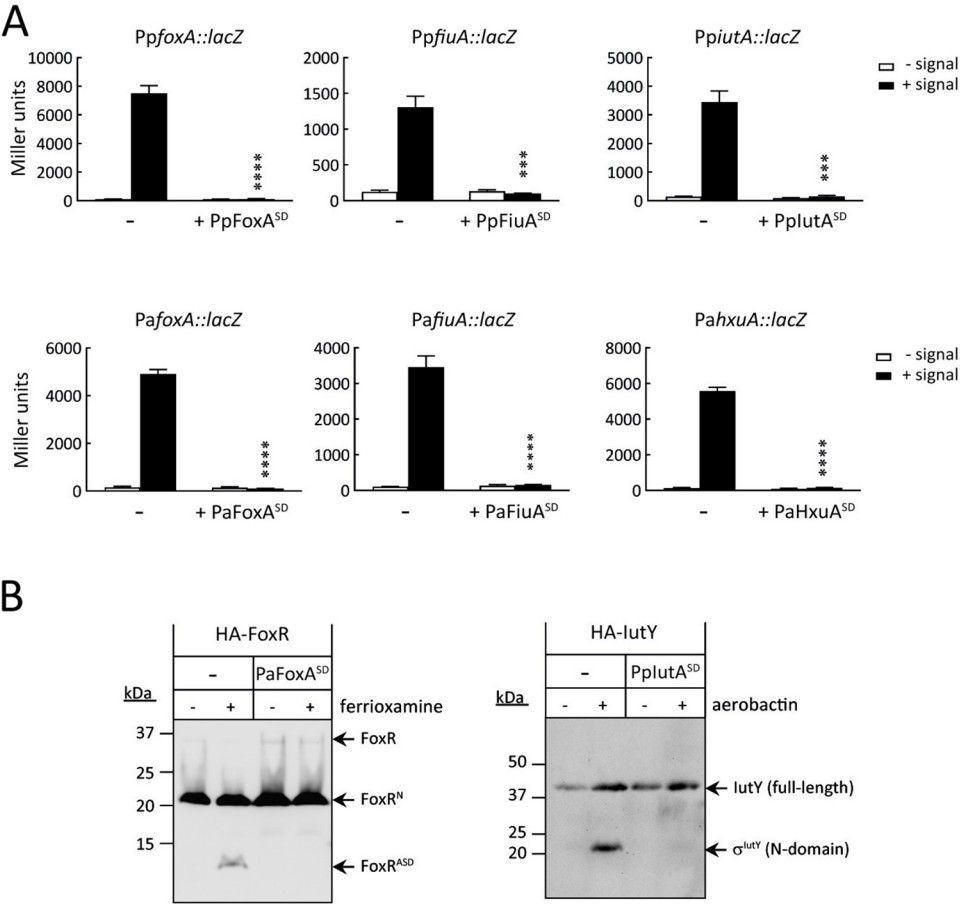

**Fig 1. Effect of overproducing the SDs of CSS receptors on CSS activity and anti-σ factor proteolysis.** In both panels, *P. putida* KT2440 or the *P. aeruginosa* PAO1 wild-type strains bear the empty pBBR1MCS-5 plasmid (-) or the pBBR1MCS-5-derived plasmid expressing the indicated SD (+ Pp- or Pa-SD) (S1 Table). Strains were grown in iron-restricted conditions (- signal) or iron-restricted medium supplemented with the cognate CSS inducing signal (+ signal), i.e., 1 μm ferrioxamine B for the Pp and Pa Fox systems, 40 μm ferrichrome for the Pp and Pa Fiu systems, aerobactin-containing supernatant for the Pp Iut system, and 20 μm haem for the Pa Hxu system. 1 mM IPTG was added to the cultures used in panel B. **(A)** β-galactosidase activity of the indicated *lacZ* fusion genes produced from pMP220-derived plasmids (S1 Table). Data are means ± SD from 3 biological replicates (*N* = 3). *P*-values were calculated by two-tailed *t* test by comparing the value obtained in the strain overexpressing the SD with that of the wild-type strain in the same growth condition and are represented in the graphs by ***, *P* < 0.001; and ****, *P* < 0.0001. **(B)** Western-blot analyses of *P. aeruginosa* HA-FoxR anti-σ factor and *P. putida* HA-IutY σ/anti-σ factor hybrid protein produced from pMMB67EH-derived plasmids (S1 Table). Proteins were immunoblotted against the HA-epitope using a monoclonal antibody. Positions of the protein fragments and the molecular size marker (in kDa) are indicated. Presence of the HA-tag adds ∼1 kDa to the molar mass of the protein fragments. Blots are representatives of at least 3 biological replicates (*N* = 3). The raw data underlying the graphs shown in the figure can be found at Mendeley Data repository (Mendeley Data, V1, 10.17632/nxh4c8ymnn.2). Western blot can be found in S1 Raw Images. CSS, cell-surface signaling; SD, signaling domain.

lane 3), corresponding to the FoxR full-length protein (approximately 36 kDa), and the 2 protein domains that are formed upon the spontaneous cleavage of FoxR, FoxR$^N$ (approximately 21 kDa), and FoxR$^C$ (approximately 15 kDa) (S1B and S2A Figs). Producing the HA-tagged proteins resulted in FoxR bands of higher intensity, and protein bands were detected also in strains not producing FoxA$^{SD}$ (S2B Fig, upper panel). However, as occurred with the non-tagged protein, co-expression of the FoxR HA-tagged proteins with FoxA$^{SD}$ considerably increased the intensity of the FoxR protein bands (S2B Fig, lanes 5 and 7). The same 3 protein

fragments detected with FoxR were detected with the HA-tagged versions (note that the HA epitope adds 1 kDa to the molecular weight of the corresponding protein band) (S2B Fig, upper panel). In addition, an extra band of approximately 32 to 33 kDa was detected with FoxR-HA (S2B Fig, lanes 6–7). This FoxR truncate is likely a consequence of the presence of the HA epitope in the C-terminal end because this band was not detected in the non-tagged or the N-terminally HA-tagged versions. Altogether, these results show that FoxA$^{SD}$ can stabilize FoxR preventing it from degradation. Importantly, this effect depends on the presence of ferrioxamine because it does not occur when the strains are grown in iron-sufficient conditions (S2C Fig). In this condition, the intensities of all FoxR protein bands were higher than the intensities of the FoxR protein bands when bacteria were grown in the presence of ferrioxamine and did not increase when FoxR was co-produced with FoxA$^{SD}$ (S2C Fig). This suggests that under iron limitation and in the presence of ferrioxamine, a cell factor destabilizes FoxR probably by proteolysis. Therefore, overproducing FoxA$^{SD}$ impedes both the regulated proteolysis of FoxR (Fig 1B) and its degradation (S2B Fig) thus blocking CSS activity (Fig 1A).

## The C-terminal domain of FoxR interacts with the signaling domain of FoxA

The experiments performed above suggest that FoxA$^{SD}$ interacts with FoxR protecting it from proteolysis. To analyze this interaction, we performed two-hybrid experiments using the bacterial adenylate cyclase-based two-hybrid (BACTH) system [30,31]. This system exploits the fact that the catalytic domain of adenylate cyclase (CyaA) from *Bordetella pertussis* consists of 2 complementary fragments, T25 and T18, that are not active when physically separated. When these 2 fragments are fused via 2 interacting polypeptides, the CyaA enzyme is functional and synthesizes cAMP, which is required to express the lactose (*lac*) operon (among other genes). The amount of cAMP produced by reconstituted CyaA can be thus measured by β-galactosidase assay using an *E. coli cyaA* negative strain [31]. Hence, we fused FoxA$^{SD}$ (amino acids 1–89 of the FoxA mature protein) to the T25 fragment of the *B. pertussis* CyaA enzyme (FoxA$^{SD}$-T25) and different FoxR domains to the T18 fragment. We included the whole periplasmic domain of FoxR containing the T192A mutation that prevents the spontaneous cleavage of this protein [22] (FoxR$^{peri-T192A}$, amino acids 107–328), and the FoxR$^{C}$ (amino acids 192–328) and FoxR$^{N}$ (amino acids 1–191) domains produced upon the spontaneous cleavage of this protein (Fig 2A). In addition, we included a FoxR fragment containing only the periplasmic part of FoxR$^{N}$ without the cytosolic and TM domains (FoxR$^{peri-N}$, amino acids 107–191, Fig 2A). These constructs were introduced into the *E. coli cyaA* negative BTH101 strain bearing the FoxA$^{SD}$-T25 construct. As a negative control, we used a strain bearing the pKT25 and pUT18C empty plasmids (-), and as a positive control, a strain encoding T25 and T18 fragments fused to the GCN4 leucine-zipper dimerization motif (T25-zip and T18-zip) that produces a characteristic Cya$^{+}$ phenotype [30] (Fig 2B). β-galactosidase assays showed that the Cya$^{+}$ phenotype was restored when FoxA$^{SD}$ was co-produced with the FoxR$^{peri-T192A}$ and the FoxR$^{C}$ domains but not with FoxR$^{N}$ or FoxR$^{peri-N}$ domains (Fig 2B). This indicates that FoxA$^{SD}$ interacts with the C-domain of FoxR but not with its N-domain.

To investigate the oligomeric states of FoxR$^{peri}$ and FoxA$^{SD}$ protein domains and the stoichiometry of the FoxR$^{peri}$/FoxA$^{SD}$ complex, we performed sedimentation velocity analytical ultracentrifugation (AUC). FoxA$^{SD}$ and 3 different FoxR domains (FoxR$^{peri-N}$, FoxR$^{C}$, and FoxR$^{peri-T192A}$, Fig 2A) were N-terminally tagged with a 6xHis epitope and heterologously produced in *E. coli* for protein purification. Analyses of the expression and solubility of the Fox domains showed that FoxR$^{C}$ and FoxR$^{peri-N}$ were highly insoluble (S4 Fig). However, FoxR$^{peri-T192A}$ containing the whole periplasmic region of FoxR was soluble, as was FoxA$^{SD}$ (S4 Fig).

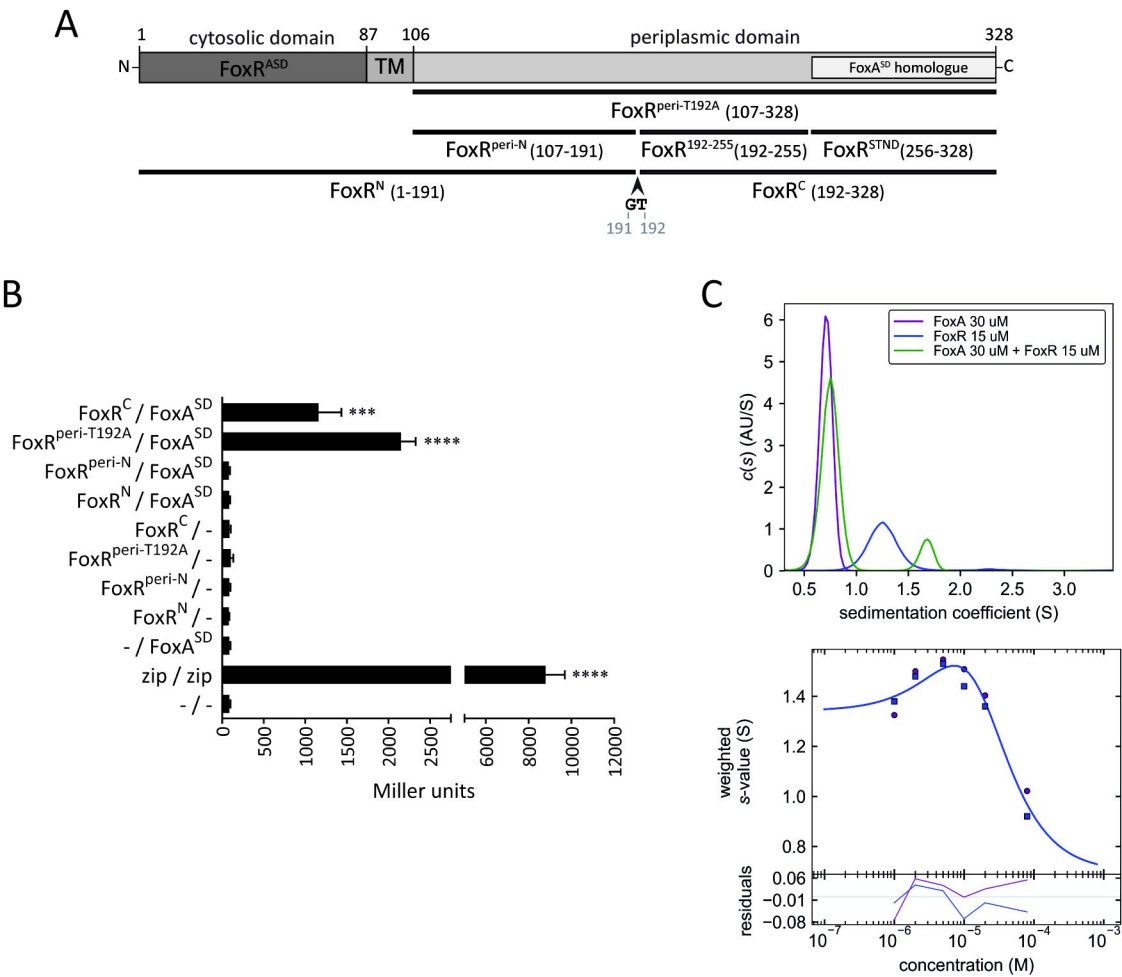

**Fig 2. Analysis of the FoxR/FoxA$^{SD}$ interaction in vivo by two-hybrid and in vitro by analytical ultracentrifugation. (A)** Schematic representation of the FoxR protein and the protein fragments used in the two-hybrid assay. The amino acids contained in each fragment are indicated in brackets. **(B)** Bacterial two-hybrid assay to test interactions between FoxA$^{SD}$ and the different FoxR fragments shown in A. *E. coli* BTH101 bearing the pUTC18C and pKNT25 empty plasmids (- /-) or its derivatives encoding the indicated FoxR and FoxA$^{SD}$ proteins were grown in LB with 0.5 mM IPTG. A strain bearing the pUTC18C-zip and pKNT25-zip plasmids (zip / zip) was used as a positive control. β-galactosidase assay was performed, and data are means ± SD from at least 3 biological replicates (*N* = 3). *P*-values were calculated by two-tailed *t* test by comparing the value obtained for one strain with that of the negative control (- / -) and are represented in the graphs by ***, $P < 0.001$; and ****, $P < 0.0001$. **(C)** Biophysical characterization of the FoxR/FoxA$^{SD}$ interaction. The upper panel shows the sedimentation velocity AUC analysis of FoxR$^{peri}$, FoxA$^{SD}$, and the FoxR$^{peri}$/FoxA$^{SD}$ complex. Values given correspond to experiments conducted at 8°C in Tris 20 mM (pH 8.3), NaCl 500 mM, DTT 1 mM, Glycerol 3% buffer and have not been standardized to s$_{20,w}$. The lower panel shows the sw isotherm analysis of the sedimentation coefficients from a FoxR$^{peri}$/FoxA$^{SD}$ titration. The solid curve indicates the best fit of a global analysis of 2 sw-isotherms (absorbance and interference data). The bottom panel shows the residual plot. The raw data underlying the graphs shown in the figure can be found at Mendeley Data repository (Mendeley Data, V1, 10.17632/nxh4c8ymnn.2). AUC, analytical ultracentrifugation;

Thus, these 2 protein domains were purified by FPLC and used in the AUC experiments. The theoretical standardized sedimentation values (s$_{20,w}$, indicating values at 20°C and in water solvent) were estimated by hydrodynamic modeling using HYDROPRO [32] on the atomic detailed structures of the monomeric proteins obtained from the AlphaFold2 database [33]. For FoxR$^{peri}$ the s$_{20,w}$ was estimated at 2.4 S and for FoxA$^{SD}$ at 1.2 S. AUC was performed first with the single protein domains, both at a concentration of 15 μm. The obtained sedimentation coefficient profiles showed an almost unique sedimenting species for both proteins under

the conditions assayed (Fig 2C, upper panel). Upon standardization of the values obtained to water and 20˚C conditions (see Materials and methods), the sedimentation coefficients of these species were 2.1 S for FoxR$^{peri}$ and 1.2 S for FoxA$^{SD}$. These values correspond to the theoretically calculated coefficients of the monomers indicating that none of the protein domains oligomerized at the concentration assayed. The assay was then performed with FoxR$^{peri}$ and FoxA$^{SD}$ in the same cell to evaluate their oligomerization states and the stoichiometry of the complex. Several concentration ratios were assayed (FoxR$^{peri}$:FoxA$^{SD}$ at 7.5 μm:15 μm, 15 μm:15 μm, and 15 μm:30 μm), showing all a similar sedimentation coefficient profile. In all cases, the sedimenting species corresponding to FoxA$^{SD}$ was present while the species corresponding to FoxR$^{peri}$ disappeared (Fig 2C, upper panel). Instead, a new species with a $s_{20,w}$ at 2.9 S appeared (Fig 2C, upper panel). This value correlates with the theoretical sedimentation coefficient of the FoxR$^{peri}$/FoxA$^{SD}$ complex (2.7 S) calculated form the hydrodynamic model generated with AlphaFold2 (Fig 3A). This indicates that FoxR$^{peri}$ and FoxA$^{SD}$ form a heterodimer complex. Finally, a series of protein concentration ratios were assayed by sedimentation velocity to estimate the binding parameters of the hetero association process. In these assays FoxR$^{peri}$ was kept fixed at 10 μm, and titrated with FoxA$^{SD}$ at 1, 2, 5, 10, 20, 40, and 80 μm. The sedimentation coefficient profile obtained was then globally analyzed, extracting the weight average $s$ for each profile of all the peaks corresponding to monomers and complex (it is a fast reaction boundary, rapid interacting system). An isotherm was generated from this data (Fig 2C, lower panel) and globally fitted through SEDPHAT [34] to an A+B <-> AB model with a Kd = 4.3 μm. The sedimentation coefficient of the larger species is consistent with the 1:1 FoxR$^{peri}$/FoxA$^{SD}$ complex.

## Structural prediction of the FoxR/FoxA$^{SD}$ interaction

Next, we aimed at determining the structure of the FoxR protein and the FoxR/FoxA$^{SD}$ complex. Therefore, highly purified FoxR$^{peri-T192A}$ and FoxA$^{SD}$ protein domains were obtained by SEC-FPLC. However, several attempts to obtain crystals of the FoxR protein alone and the FoxR/FoxA$^{SD}$ complex were unsuccessful. Therefore, we obtained structural predictions for these proteins. The structural model of the full-length FoxR protein was obtained from the Alphafold structure database (AF_AFQ9I115F1). As predicted from the amino acid sequence, FoxR has an N-terminal ASD spanning residues 1–80 that display an interlocked three-helix bundle followed by a flexible loop that connects it to the transmembrane region (residues 87–106) (S5A Fig). Two structural subdomains conforming the periplasmic portion of the protein are evident in the structural model (S5A Fig). The N-terminal subdomain spans residues 107–247 and is composed of 14 anti-parallel β-strands forming a twisted β-solenoid-like motif, while the C-terminal subdomain (residues 248–328) folds into an STN-like domain (STND) (S5A Fig). The structure of the periplasmic domain of FoxR is nearly identical to that of the crystal structure of anti-σ factor PupR (6OVM), with a root mean square deviation (RMSD) of 0.867 Å (S5B Fig). The most significant dissimilarities are in the 2 anti-parallel β strands at the beginning of the STND (S5C Fig).

As mentioned before, the C-terminal subdomain of the periplasmic region of FoxR (FoxR$^{STND}$) is a structural homologue of the FoxA$^{SD}$ (PBD access code 6I97). Even though both domains share the same fold, they superimpose with an RMSD of 1.709 Å, denoting a significant number of structural differences (S6C Fig). Both domains share the same structural elements, although FoxR$^{STND}$ presents an extra 5 amino acid α helix at the N-terminus (α1), followed by a common structure composed of 6 β strands and 3 α helices (α2, α3, and α4). Strands β1 and β3 form antiparallel β sheets at the N-terminus, the remaining β strands β2, β5, and β6 form C-terminal β sheets (S6A Fig). While lacking the first α helix, FoxA$^{SD}$ is similarly

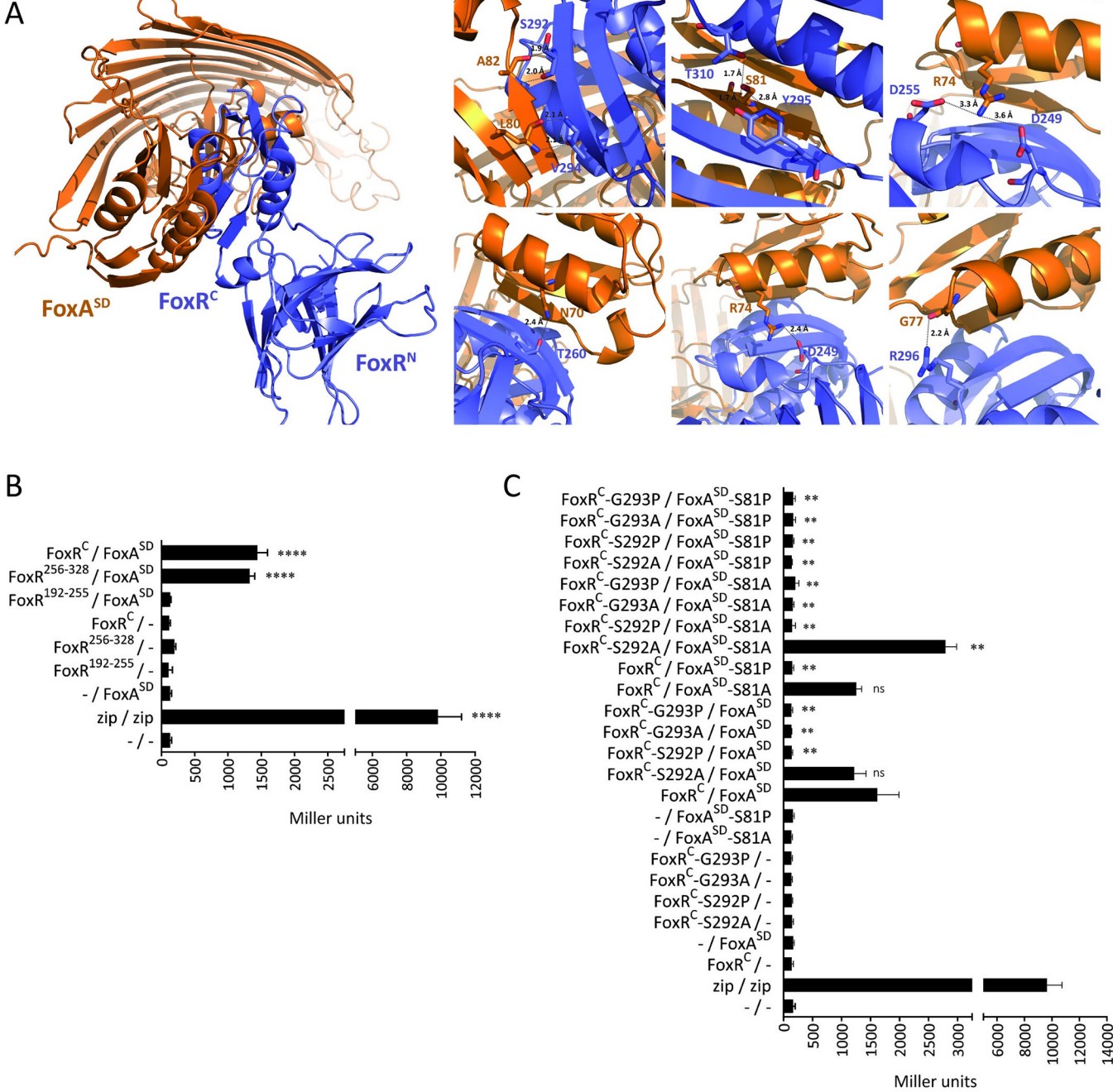

**Fig 3. Structural prediction of the FoxR/FoxA interaction and effect of key residues on FoxR/FoxA interaction. (A)** The signaling domain of FoxA (in orange) interacts with FoxR (in blue) via their STN-like domains. Close-up view involving the residues predicted to participate in the FoxR/FoxA interaction and the distances between the interacting residues are shown. The interaction model was generated with Alphafold2 using the FoxR[peri] domain (amino acids 107–328, Fig 2A) and FoxA[1-516]. **(B, C)** Bacterial two-hybrid assay to test interactions of the 2 subdomains of FoxR[C] with FoxA[SD] **(B)**, and between the FoxR[C] and FoxA[SD] protein variants bearing the indicated single residue change **(C)**. *E. coli* BTH101 bearing the pUTC18C and pKNT25 empty plasmids (- /-) or its derivatives encoding the indicated FoxR[C] and FoxA[SD] proteins were grown in LB with 0.5 mM IPTG. A strain bearing the pUTC18C-zip and pKNT25-zip plasmids (zip / zip) was used as a positive control. β-galactosidase assay was performed, and data are means ± SD from 3 (*N* = 3) biological replicates. *P*-values were calculated by two-tailed *t* test by comparing the value obtained in the strain bearing the changed protein variant(s) with that of the strain bearing the wild-type FoxR[C] / FoxA[SD] constructs and are represented in the graphs by ns, no significant; **, *P* < 0.01; ***, *P* < 0.001; and ****, *P* < 0.0001. The raw data underlying the graphs shown in the figure can be found at Mendeley Data repository (Mendeley Data, V1, 10.17632/nxh4c8ymnn.2).

composed of an N-terminal β sheet (β1' and β3'), 3 α helices (α1', α2', and α3'), and the C-terminal β sheet (β3', β4', and β5') (S6B Fig).

The interaction model between FoxR[peri] and FoxA we generated with Alphafold shows that both proteins are likely to interact through their STN domains with a predicted alignment error (PAE) <10 Å along their STN domains (Figs 3A and S7). This agrees with the oligomeric nature of STN domains [13,14]. According to this prediction, the FoxR/FoxA[SD] complex forms a 5-stranded β sheet composed of β1 and β3 from FoxR[STND], and β2', β5', and β6' from FoxA[SD]. This β sheet interface is created by 4 hydrogen bonds between the backbones of the β3 residues V294 and S292 from FoxR[STND] and L80 and A82 in β2' from FoxA (Fig 3A). This complex is further stabilized by 3 hydrogen bonds connecting the side chains of FoxA-S81 with residues Y295 and T310 of FoxR, which may be important for the specificity between both proteins since this is not a conserved residue in other TBDT proteins. Two salt bridges linking FoxR-D249 and -D255 to FoxA-R74, as well as 3 hydrogen bonds between FoxR residues T260, D249, and R296, and FoxA residues N70, R74, and G77, also seem to contribute to stabilizing the protein complex (Fig 3A). Many of these interactions are mirroring those observed for the crystal structure for the PupB/PupR complex [16], which supports the prediction of the interaction model (S7A Fig). In this complex, the backbone interactions between PupR-T288 and -S286 and PupB-L75 and -T77 shape the β sheet interface and the side chain of PupB-S76 stabilizes the interaction via 2 hydrogen bonds with PupR-A300 and -T304.

## FoxR[STND] is required for interactions with FoxA[SD]

The structural modeling data suggested that FoxR[STND] is the subdomain involved in interacting with FoxA[SD] (Fig 3A). To analyze the importance of this subdomain of FoxR in the FoxR/FoxA[SD] interaction, we performed BACHT 2 hybrid analyses using a FoxR[C] fragment lacking the STND subdomain and a fragment containing only this region of FoxR (FoxR[192-255] and FoxR[STND] amino acids 256–328, respectively, Fig 2A). As shown before, β-galactosidase activity increased when FoxA[SD] was co-expressed with the whole FoxR[C] domain (residues 192–328) (Fig 3B). Activity also increased when FoxA[SD] was co-expressed with FoxR[STND], but it did not when it was co-expressed with FoxR[192-255] (Fig 3B). This indicates that FoxR[192-255] does not interact with FoxA[SD] and thus that the interaction between FoxR and FoxA[SD] requires the C-terminal subdomain of FoxR homologous to FoxA[SD]. This agrees with the structural prediction showing that the STND subdomain of FoxR interacts with FoxA[SD] by forming a 5-stranded β sheet stabilized by α helices.

## Residues FoxR-S292 and -G293, and FoxA-S81 are required for FoxR[C]/FoxA[SD] interaction and CSS signaling

Because the Alphafold model of the FoxR/FoxA complex indicates that the region comprising residues S292 to V294 of FoxR[STND] is required to create a β sheet interface, and the residue S81 of FoxA is required to stabilize this interface (Fig 3A), we decided to change these protein residues and analyze the FoxR/FoxA interaction by two-hybrid experiments. We selected the FoxR[C]-S292 and -G293, and FoxA[SD]-S81 residues, and changed them to alanine (A) and proline (P). The FoxR-S292P change is expected to distort the β3 strand required to interact with FoxA, while the S292A change should not introduce significant structural alterations. Besides, due to the structural features of glycines, changing the FoxR-G293 residue may alter the proper formation of the β3 strand, also perturbing its interaction with FoxA. As for FoxA, a S81A change will impair the hydrogen bond interactions observed with FoxR-Y295 and -T310 residues without perturbing the overall structure of the β2' strand, while a S81P change will impair both the hydrogen bonds and the β2' strand formation. Our two-hybrid analyses using these

constructs showed that none of the β3-distorting variants of FoxR were able to interact with FoxA$^{SD}$ (Fig 3C). Similarly, the β2'-distorting FoxA$^{SD}$-S81P variant was not able to interact with FoxR$^C$. However, the 2 variants that did not distort the secondary β structure of the proteins, the FoxA$^{SD}$-S81A and FoxR$^C$-S292A variants, respectively, were able to interact with their partners to almost wild-type levels (Fig 3C). In fact, for FoxA$^{SD}$-S81A, we did not observe any significant decrease in binding despite the loss of the 3 hydrogen bonds in the interaction with FoxR$^C$ (Fig 3A). However, the interaction between FoxA$^{SD}$-S81A and FoxR$^C$-S292A seems to be stronger than the interaction between wild-type FoxR$^C$ and FoxA$^{SD}$ (Fig 3C) hinting at a more promiscuous binding towards alternative variants of FoxR$^C$. Altogether, this suggests that the 3 hydrogen bonds coordinated by FoxA-S81 might be involved in the stabilization and/or specificity of the complex between both proteins, rather than being essential for complex formation.

Impairing the FoxR/FoxA interaction may impact the proteolytic cascade and CSS activity. To analyze this effect, we overproduced the FoxA$^{SD}$-S81A and -S81P variants in the *P. aeruginosa* PAO1 wild-type strain and analyzed σ$^{FoxI}$ activity (Fig 4A) and FoxR proteolysis (Fig 4B). Overexpression of the FoxA$^{SD}$-S81A variant blocks CSS activation and FoxR proteolysis in response to ferrioxamine, as also observed with the FoxA$^{SD}$ wild-type (Fig 4A and 4B). In contrast, overproduction of the FoxA$^{SD}$-S81P variant reduced but did not block σ$^{FoxI}$ activity and FoxR proteolysis (Fig 4A and 4B). Because FoxA$^{SD}$-S81A is still able to interact with FoxR while FoxA$^{SD}$-S81P is not (Fig 3C), these results indicate that the ability of FoxA$^{SD}$ to protect FoxR from proteolysis and block CSS activity depends on its capacity to interact with the anti-σ factor.

## The signaling domain of FoxA (FoxA$^{SD}$) is cleaved

Our earlier results indicate that the periplasmic proteases Prc and CtpA are required for proper CSS activation. While lack of Prc considerably reduces CSS activity in response to the inducing signal, lack of CtpA significantly increases CSS activity [23]. Absence of both proteases influence the proteolysis of the anti-σ factor FoxR but in opposite ways. While in a Δ*prc* mutant the FoxR$^C$ domain is more stable, in accordance with Prc degrading this domain, in a Δ*ctpA* mutant FoxR$^C$ is hardly detectable [23]. These phenotypes may be related to FoxA, as we have determined in this study that this receptor protects FoxR from degradation. Therefore, we decided to investigate the effect that the Prc and CtpA proteases have on FoxA. Stability of FoxA was analyzed by western blot in the PAO1 wild-type strain and the Δ*ctpA* and Δ*prc* mutants using a polyclonal antibody generated against a peptide of the β-barrel of the receptor (residues 331–344). A mutant unable to produce FoxA (*foxA*-Tn) was used as negative control. As expected, the FoxA receptor was not detected when ferrioxamine was not present in the medium while it was highly induced when the siderophore was added (Fig 4C). In this condition, a single protein band was observed in the PAO1 and Δ*ctpA* mutants; however, 2 protein bands were obtained in the Δ*prc* mutant (Fig 4C). Interestingly, the sizes of these protein bands resemble the size of the complete FoxA protein (~84 to 85 kDa) and of the FoxA protein without the signaling domain (~73 to 74 kDa). This suggest that not only FoxR but also FoxA is processed in vivo in the periplasm and that this event is required for proper CSS activity.

## Discussion

In this study, we shed light on the molecular details of the signal transduction by CSS by analyzing the interaction between the TBDT and the anti-σ factor. We show that overproducing the SD of different TBDTs blocks the activation of the CSS pathway (Fig 1A) by preventing the proteolysis of the CSS anti-σ factor (Fig 1B) and thus the release of the σ$^{ECF}$ factor. This

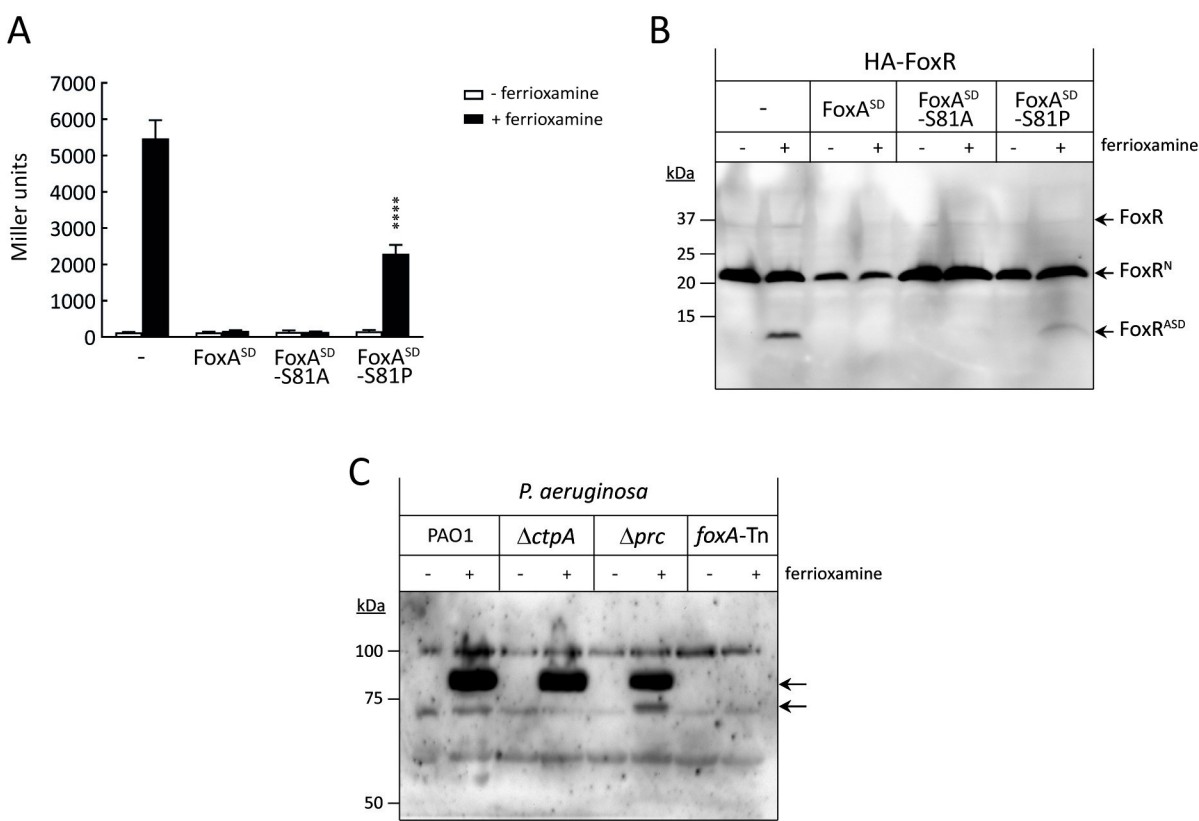

**Fig 4. Role of FoxA$^{SD}$ in CSS activity. (A)** β-galactosidase activity of the *P. aeruginosa foxA::lacZ* fusion gene in the PAO1 wild-type bearing the empty pBBR1MCS-5 plasmid (-) or the pBBR1MCS-5-derived plasmid expressing the wild-type FoxA$^{SD}$ protein or the indicated protein variant (S1 Table). Strains were grown in iron-restricted conditions without or with 1 μm ferrioxamine B. Data are means ± SD from 5 ($N$ = 5) biological replicates. *P*-values were calculated by two-tailed *t* test by comparing the value obtained in the strain overproducing the single change residue protein variant with that of the wild-type FoxA$^{SD}$ protein in the same growth condition and are represented in the graphs by ****, $P < 0.0001$. **(B)** Western blot analyses of *P. aeruginosa* PAO1 producing an N-terminally HA-tagged FoxR anti-σ factor from a pMMB67EH-derived plasmid (S1 Table). Strains also bear the pBBR1MCS-5 empty plasmid (-) or the pBBR1MCS-5-derived plasmid expressing the wild-type FoxA$^{SD}$ protein or the indicated protein variants (S1 Table) and were grown in iron-restricted conditions without (-) or with (+) 1 μm ferrioxamine B and 1 mM IPTG. Proteins were immunoblotted against the HA-epitope using a monoclonal antibody. Positions of the protein fragments and the molecular size marker (in kDa) are indicated. Presence of the HA-tag adds ∼1 kDa to the molar mass of the protein fragments. Blots are representatives of 3 biological replicates ($N$ = 3). **(C)** *P. aeruginosa* PAO1 wild-type strain and the indicated isogenic mutants were grown to late log-phase under iron-restricted conditions and in the absence (-) or presence (+) of 1 μm ferrioxamine. Total proteins were prepared and immunoblotted against the FoxA TBDT using a polyclonal antibody directed against a peptide of the β-barrel of the protein. Positions of the protein fragments and the molecular size marker (in kDa) are indicated. Blot is a representative of 4 biological replicates ($N$ = 4). The raw data underlying the graphs shown in the figure can be found at Mendeley Data repository (Mendeley Data, V1, 10.17632/nxh4c8ymnn.2). Western blot can be found in S1 Raw Images. CSS, cell-surface signaling; TBDT, TonB-dependent transducer.

indicates that overproducing the SD of the receptor makes the anti-σ factor inaccessible to the proteases required to degrade and cleave this protein. In accordance with this, we show that the FoxR anti-σ factor is considerably more stable upon overproduction of FoxA$^{SD}$ (S2 Fig). In fact, when purifying FoxR$^{peri}$ for the in vitro analyses, we observed that a high fraction of the protein precipitated upon dialysis into low-salt buffers or when concentrating it, while no precipitation was observed when the protein was dialyzed in presence of FoxA$^{SD}$. This further confirms that FoxA stabilizes the periplasmic domain of FoxR, as also observed with the PupB TBDT and the PupR anti-σ factor [16]. Based on these results, we hypothesized that in vivo, FoxA$^{SD}$ interacts with FoxR in the absence of ferrioxamine, and that this interaction stabilizes and blocks the proteolysis of the anti-σ factor.

Three forms of the FoxR protein are detected in *P. aeruginosa*, the full-length protein (~36 kDa), the FoxR$^N$ domain (~21 kDa), and the FoxR$^C$ domain (~15 kDa) (S2 Fig). FoxR$^N$ and FoxR$^C$ are the result of the non-enzymatic self-cleavage of FoxR, which occurs prior to signal recognition by an NO-acyl shift between GT residues located within the periplasmic domain of FoxR [22,28]. We showed earlier that FoxR$^N$ and FoxR$^C$ interact with each other and function together in the signal transduction pathway [22]. FoxR$^C$, but not FoxR$^N$, is able to sense and generate the response to ferrioxamine, suggesting an interaction between FoxR$^C$ and FoxA [22]. The two-hybrid analyses performed in this work have demonstrated the interaction of FoxA$^{SD}$ with FoxR$^C$ but not with FoxR$^N$ (Fig 2B). Specifically, FoxA$^{SD}$ interacts with the subdomain comprising residues 256–328 of FoxR$^C$, as shown by both the two-hybrid assay and the FoxR/FoxA predicted structure (Fig 3B and 3A). This domain is a structural homologue of the periplasmic domain of secretins and the signaling domain of TBDTs (the SNT domain). Secretins form megadalton bacterial membrane channels by assembling 12–15 monomers. Bacteria use these channels to transport large molecules across the outer membrane, including effectors of the type 2 secretion system (T2SS), filament accommodation of the type 3 secretion and type 4 (pili) systems (T3SS and T4P), or phage release in the filamentous phage extrusion (FPE) system [13,14]. Secretins contain a C-terminal domain that forms the channel in the outer membrane and a variable number of periplasmic STN-like domains connected via short linkers. The STNDs of one secretin monomer interact with both each other and the STNDs from other secretin molecules forming oligomers. Due to the similarities with secretin domains, one might hypothesize that the SD of TBDTs and the SNT domain of anti-σ factors form similar complexes in vivo. In this regard, it is noteworthy to mention that when FoxA$^{SD}$ was overproduced, we detected protein bands of higher molecular weight than that of the FoxA$^{SD}$ monomer (9 to 10 kDa) (S2B and S2C Fig, middle panel). These protein bands could be the result of FoxA$^{SD}$ forming dimers or trimers. However, in our AUC experiments, FoxA$^{SD}$ was present as a monomer (Fig 2C). This difference could be related to the presence of an N-terminal His-tag in the FoxA$^{SD}$ protein used in the in vitro experiments, which was not present in the construct we used for the experiments in vivo. As such, it might be possible that the N-terminal His-tag inhibits the dimerization of FoxA$^{SD}$. Another possibility would be the formation of a heterodimer between FoxA$^{SD}$ and FoxR$^C$ similarly to the secretin complexes. Moreover, it has been shown that in the periplasm SNT domains can interact with cytoplasmic membrane proteins through high homology regions (HR) located next to C-terminal PDZ domains within the cytoplasmic membrane proteins [35,36]. Interestingly, PDZ domains are present in several proteases, including the periplasmic proteases Prc and CtpA involved in activating CSS pathways [23,37]. Hence, one could suggest that the SD of TBDTs not only interact with the periplasmic domain of the anti-σ factor but also with proteases. In the same way, the SNT-like domain of the anti-σ factor could interact with proteases. In this regard, it is important to mention that, while the absence of Prc in a null mutant diminishes but not abolishes CSS activation, a proteolytically inactivated version of this protease exhibited a dominant negative effect and completely inhibited CSS activation [23]. This is likely the result of tight binding, but not cleavage, of Prc to the anti-σ factor or to another component of the CSS system, which agrees with the formation of a FoxR$^C$/FoxA$^{SD}$/Prc complex. Additionally, the region downstream of the SD of FoxA interacts with TonB [10], an interaction that is required both for transporting the siderophore and for signaling. So, one might imagine the formation of different complexes in the periplasm regulating signaling and the transport of the siderophore through TBDTs.

Formation of an anti-σ factor/SD complex in the absence of the inducing signal has also been proposed by Jensen and colleagues after solving the structure of the PupR/PupB$^{SD}$ complex [16]. Because our attempts to obtain crystals of the FoxR/FoxA$^{SD}$ complex were

unsuccessful, we generated a structural prediction of this complex (Fig 3A). Most of the interactions detected in the FoxR/FoxA complex mirror those observed in the crystal structure of the PupR/PupB complex [16], which involve the STN-like domains of both proteins. By changing residues involved in the FoxR/FoxA interaction, we showed that disrupting any of the β strands conforming the β-sheet interface leads to a defective interaction between both proteins. However, disrupting the 3 hydrogen bonds stablished by the FoxA-S81 residue with FoxR-Y295 and -T310 did not impair the formation of the FoxR/FoxA complex (Fig 3C, FoxA$^{SD}$-S81A). Importantly, while the interacting FoxA$^{SD}$-S81A variant can block the proteolysis FoxR, the non-interacting FoxA$^{SD}$-S81P protein cannot (Fig 4A and 4B). This indicates that the interaction with FoxR is essential for FoxA$^{SD}$ to be able to protect the anti-σ factor from proteolysis, which further suggests that FoxR and FoxA already interact in the absence of the inducing signal.

A constitutive complex in the absence of ferrioxamine is also formed between the FoxA receptor and the TonB protein [10]. This constitutive FoxA/TonB interaction is mediated by an unstructured polypeptide segment located upstream the TonB box of FoxA (residues 135–141). This segment is only present in TonB-dependent receptors involved in signaling which contain the additional N-terminal domain that forms the SD. Although the biological function of the TonB/FoxA constitutive interaction is still unknown, it has been proposed that it may be related to the signaling function of FoxA [10]. Binding of ferrioxamine to FoxA triggers conformational changes in extracellular loops of the protein but also in the periplasmic region provoking the expulsion of the TonB box from inside the barrel. This allows the formation of a high-affinity FoxA/TonB complex by β-augmentation, an interaction that is strengthened by complementary contacts between the unstructured polypeptide segment of FoxA and the surface of TonB [10]. Formation of this high-affinity FoxA/TonB complex triggers the signaling domain of FoxA to rotate but does not alter the overall structure of this domain [10]. Importantly, this displacement does not imply that FoxA$^{SD}$ is further extended into the periplasm, as observed for other TBDTs [20]. Therefore, it is unclear how FoxA$^{SD}$ can span that far across the periplasm to interact with FoxR$^C$ and link the cytoplasmic and the outer membranes. The solution to this problem may be related to our finding that FoxA$^{SD}$ is cleaved, as we propose based on the detection of 2 different FoxA forms (Fig 4C, Δ$prc$ strain). Based on our previous results, we believe that the protease performing this cleavage is CtpA. We showed earlier that CtpA is required for proper CSS activity, and that this protease does not process the FoxR anti-σ factor but another component of the CSS pathway [23]. The phenotype of a $ctpA$ mutant is in accordance with FoxA$^{SD}$ not being cleaved because the FoxR$^C$ domain is very unstable in this mutant and CSS activity is considerably higher than in the wild-type strain ([23], see Figs 3B and 2A of this article). Moreover, we know that CtpA works upstream of Prc in the signaling cascade because a double $prc$ $ctpA$ mutant has the same phenotype as the single $prc$ mutant [23]. Taking all these results into account, we propose that the SD of FoxA is cleaved, and that this cleavage is performed by the CtpA protease. We believe that, in the wild-type strain, the cleavage of FoxA$^{SD}$ occurs in the absence of ferrioxamine, resulting in the stabilization of the FoxR$^C$ domain, which in turn blocks the RIP of the FoxR$^N$ domain and thus the release of σ$^{FoxI}$ (S1B Fig). However, due to low amounts of the FoxA protein in the absence of ferrioxamine (Fig 4C), it has not been possible to detect this proteolytic event in the wild-type strain and further analysis is required to confirm this prediction. In the presence of ferrioxamine, rotation of the signaling domain of FoxA [10] would prevent the cleavage and release of FoxA$^{SD}$ thus allowing the access of Prc to FoxR$^C$. However, if the Prc protease is absent, the cleavage of FoxA$^{SD}$ can occur in this condition (Fig 4C, Δ$prc$). It is possible that Prc modulates CtpA levels or the access of CtpA to its substrate. Further experiments will be conducted to clarify this. Nevertheless, cleavage of the SD seems to be related to the signaling activity of

TBDTs. In this context, it is important to note that, although signaling and transport are both substrate- and TonB-dependent processes, these functions are separable, and the SD of the TBDT can be removed without affecting transport [38] and transport can be blocked still allowing signaling [39]. Therefore, separation of FoxA[SD] from the rest of the protein would still allow binding and transport of ferrioxamine.

In summary, we show in this work that FoxA and FoxR interact via their SNT-like domains, and that this interaction is required to protect the anti-σ factor from proteolysis. Furthermore, we propose that signal transduction via CSS involves the cleavage of the SD of FoxA, probably by the CtpA protease, since lack of this enzyme results in full-size FoxA protein and destabilization of FoxR. This evidence supports a novel model for CSS signaling in which in the absence of the signal, the SD of the TBDT is cleaved and protects the anti-σ from degradation under non-inducing conditions, providing an explanation for how these 2 proteins cover the physical gap between the cytoplasmic and outer membrane in the signal transduction pathway.

## Materials and methods

### Bacterial growth conditions

Bacteria were routinely grown in liquid LB [40] on a rotatory shaker at 37˚C (*P. aeruginosa* and *E. coli*) or 30˚C (*P. putida*) and 200 rpm. For iron-restricted conditions cells were cultured in CAS medium [27] containing 400 μm (for *P. aeruginosa*) or 200 μm (for *P. putida*) of 2,2′-bipyridyl. For induction experiments, the iron-restricted medium was supplemented with 1 μm iron-free ferrioxamine B (Merck), 40 μm iron-free ferrichrome (Santa Cruz Biotechnology), 20 μm haem (Merck), or aerobactin-containing supernatant in 1:1 proportion as described before [21]. Iron-rich conditions were obtained by adding 50 to 100 μm $FeCl_3$ to the iron-restricted medium. When required, 1 mM isopropyl β-D-1-thiogalactopyranoside (IPTG) was added to the medium to induce full expression from the pMMB67EH P*tac* promoter. Antibiotics were used at the following final concentrations (μg ml$^{-1}$): ampicillin (Ap), 100; gentamycin (Gm), 10; kanamycin (Km), 50; piperacillin (Pip), 25; streptomycin (Sm), 100; tetracycline (Tc), 12.5 for *E. coli* and 20 for *Pseudomonas*.

### Plasmid construction and molecular biology

Plasmids used are described in S1 Table and primers listed in S2 Table. PCR amplifications were performed using Phusion Hot Start High-Fidelity DNA Polymerase (Finnzymes) or Expand High Fidelity DNA polymerase (Roche). All constructs were confirmed by DNA sequencing and transferred to *Pseudomonas* by electroporation [41].

### Bacterial strains and mutant construction

Strains used are listed in S1 Table. The *P. aeruginosa* Δ*foxR* deletion mutant was constructed by allelic exchange using the suicide vector pKNG101 [42]. Briefly, a 500-bp DNA fragment of the 5′ (up) and 3′ (down) ends of the *foxR* gene were obtained by PCR using PAO1 chromosomal DNA as a template with proper oligonucleotides (S2 Table). The gene fragment containing the *foxR* deletion was cloned into pKNG101 (S1 Table). The pKNG101-derivatives were maintained in *E. coli* CC118λpir and mobilized into *P. aeruginosa* PAO1 by triparental mating using the *E. coli* 1047 (pRK2013) helper strain [43]. Clones in which a double recombination event occurred after counterselection on sucrose plates were selected. The *foxR* chromosomal deletion was verified by PCR.

## Enzyme assay

β-galactosidase activities in soluble cell extracts were determined using o-nitrophenyl-b-D-galactopyranoside (ONPG) (Merck) as described before [27]. Each condition was tested in duplicate in at least 3 biologically independent experiments and the data given are the average with error bars representing standard deviation (SD). Activity is expressed in Miller units.

## Protein purification

His-tagged proteins were produced in *E. coli* BL21 from pET28(+)-derivative plasmids and purified by affinity chromatography. Cells were grown in 2 L Erlenmeyer flasks containing 500 ml LB medium supplemented with 50 μg ml–1 kanamycin at 30˚C until an OD660 of 0.6, at which point protein production was induced by adding 0.1 mM IPTG. Growth was continued at 18˚C overnight prior to cell harvest by centrifugation at 10.000 × g for 30 min. Cell pellets were resuspended in 30 ml of buffer A (20 mM Tris-HCl, 0.1 mM EDTA, 300 mM NaCl, 5% (v/v) glycerol, 10 mM imidazole, 5 mM β-mercaptoethanol; pH 8.3) supplemented with 1× Complete protease inhibitor cocktail (Roche) and broken by sonication 5 times for 45 s. Following centrifugation at 20.000 × g, 4˚C for 1 h the soluble fraction was passed through a 0.22 μm filter (Millipore) and loaded onto a 5 ml HisTrapHP chelating column (GE Healthcare) previously equilibrated in buffer A. Proteins were eluted with a 10 mM to 500 mM imidazole gradient in buffer A and dialyzed against buffer B (50 mM Tris-HCl (pH 7.5), 10 mM MgCl2, 1 mM DTT).

## Production of FoxR and FoxA antibodies

FoxRperi-T192A was purified as described above and sent to Innovagen (Sweden) for antibody production. Rabbits were immunized at day 0 and subsequently given boosters at days 14, 28, 49, and 70. At day 84, rabbits were killed and serum was isolated. Prior western blot, serum was concentrated using 30K centrifugal filter units (Millipore) at 4,000 rpm for 15 min. The FoxA antibody was produced by immunizing rabbits with a synthetic 14-amino acid peptide SDTQFDHVKEERYA (amino acids 331–344) in Abyntek (Spain). This peptide was designed from a predicted highly antigenic sequence of the protein. To increase antigenicity, the peptide was coupled to the carrier protein keyhole limpet hemocyanin (KLH). To remove nonspecific antibodies, the final antibody preparation was purified by affinity chromatography with the synthetic peptide used in the immunization protocol.

## SDS-PAGE and western blot

Bacteria were grown until late log phase and pelleted by centrifugation. Samples were normalized according to the OD660 of the culture, solubilized in Laemmli buffer and heated for 10 min at 95˚C. Proteins were separated by SDS-PAGE containing 12% or 15% (w/v) acrylamide and electrotransferred to nitrocellulose membranes. Ponceau S staining was performed as a loading control. Immunodetection was realized using a monoclonal antibody directed against the influenza hemagglutinin epitope (1:1,000) (HA.11, Covance) or polyclonal antibodies directed against FoxRperi (1:500) and FoxA (1:1,000). Detection of the OprL outer membrane lipoprotein with the monoclonal MA1-6 antibody [44] was used as an extra loading control. The second antibody, either the horseradish peroxidase-conjugated rabbit anti-mouse (DAKO) or the horseradish peroxidase-conjugated goat anti-rabbit IgG (Merck), was detected using SuperSignal West Chemiluminescent Substrates (Thermo Scientific). Blots were scanned and analyzed using the Quantity One version 4.6.7 (Bio-Rad).

## Analytical ultracentrifugation

Purified proteins were dialysed against a buffer containing 20 mM Tris, 10% glycerol (vol/vol), 500 mM NaCl, 2 mM β-mercaptoethanol, pH 8.3, and injected onto sepharose column and subjected to size exclusion chromatography. Harvested fractions were dialysed against AUC buffer (20 mM Tris, 3% glycerol (vol/vol), 500 mM NaCl, 1 mM DTT, pH 8.3) and normalized to indicated protein concentrations. Experiments were performed with a Beckman Coulter Proteomelab XL-I analytical ultracentrifuge equipped with a Ti-50 rotor and double-sector cells. The dialysis buffer was used as reference in the cell. The rotor speed was set to 48,000 rpm. Rayleigh interference optics and absorbance optics at λ = 228 nm were used to monitor the sedimentation profiles. The temperature was set to 8°C. Data were collected at time intervals of 1 min for a total duration of the assays of 15 h. Scans were analyzed using Sedfit software (version 16.1c) [45]. Sedimentation coefficient (s-value) distributions were obtained from sedimentation velocity experiments using the c(s) method with Simplex algorithm fitting and Tikhonov-Philips regularization options. Density and viscosity of the solvent, and partial specific volume of the proteins (vbar) were calculated via Sednterp [46]. Sedimentation coefficient values given in the text were standardized to water and 20°C conditions using the calculated viscosity and density of the solvent at the conditions of the experiment. After fitting, data were exported to GUSSI v. 1.3.2 [47] for the preparation of figures. Hydrodynamic models, for the calculation of the theoretical sedimentation coefficient of the different oligomeric species (monomers and hetero-dimer), were generated using the software HHYDROPRO [32] with AlphaFold2 [33] as the source of the structural models with a mean confidence of over 80% for the best one. Best fit to isotherms of weight-average sedimentation coefficients was done by SEDPHAT version 15-2b [34], from a global analysis of both absorbance and interference data.

## Computer-assisted analyses

Sequence analyses of the *Pseudomonas* genomes were performed at http://www.pseudomonas.com [48] and sequence alignments with ClustalW [49]. Protein tertiary structure and interaction models were obtained using Alphafold2 and Colabfold servers [33,50] and visualized using PyMOL (The PyMOL Molecular Graphics System, Version 1.8 Schrödinger, LLC). The statistical analyses described in this work were conducted using Prism 7.0 software (GraphPad).

## Supporting information

**S1 Fig. The *P. aeruginosa* Fox CSS system. (A)** Structure of the FoxA TBDT. The 22-stranded antiparallel β-barrel domain of FoxA is shown in blue and the plug domain occluding the pore in yellow. The signaling domain (SD) is shown in red and the TonB box in orange. Structure was solved in [10] and was downloaded from Protein Data Bank (PDB code 6I97). **(B)** Schematic representation of the Fox system. The 3 components of the CSS system—receptor (FoxA), anti-σ factor (FoxR), and σ$^{ECF}$ (σ$^{FoxI}$)—are shown as well as the TonB-ExbBD complex and the proteases involved in the proteolysis of FoxR. Upon synthesis, the *P. aeruginosa* anti-σ factor FoxR undergoes a spontaneous cleavage that produces 2 functional N- and C-domains (FoxR$^N$ and FoxR$^C$) that interact with each other in the periplasm and are both required for proper function. The FoxA receptor interacts with the TonB protein, which enables the energy coupled uptake of the siderophore ferrioxamine, and with the FoxR$^C$ domain via its signaling domain (FoxA$^{SD}$, red ball). In response to ferrioxamine, FoxR$^C$ is degraded by the C-terminal periplasmic protease Prc, and this event triggers the regulated intramembrane proteolysis (RIP) of the FoxR$^N$ domain by the action of (at least) 2 proteases: a (still unidentified) site-1

protease (S1P) and the site-2 RseP protease (S2P). This results in the release of $\sigma^{FoxI}$ into the cytoplasm bound to the anti-$\sigma$ domain of FoxR (FoxR$^{ASD}$). Although not experimentally demonstrated yet, FoxR$^{ASD}$ likely forms part of the transcription complex. Among other genes, $\sigma^{FoxI}$ promotes the transcription of the *foxA* receptor gene. OM, outer membrane; CM, cytoplasmic membrane; RNAP, RNA polymerase.
(TIF)

**S2 Fig. Effect of overproducing FoxA$^{SD}$ on FoxR stability. (A)** Scheme of the *P. aeruginosa* FoxR protein. The cytosolic, transmembrane, and periplasmic regions of the protein are shown. The numbers indicate amino acid positions. The site at which the self-cleavage of FoxR occurs (between Gly-191 and Thr-192) is indicated. The N- and C-domains resulting from self-cleavage are illustrated. **(B, C)** Western blot analyses of the *P. aeruginosa* FoxR protein in cells overproducing FoxA$^{SD}$. The *P. aeruginosa* Δ*foxR* mutant containing the pMMB67EH empty plasmid (-) or its derivate expressing FoxR, HA-FoxR, or FoxR-HA, and the pBBR1MCS-5 empty plasmid (-) or its derivative expressing FoxA$^{SD}$ were grown in iron-restricted medium supplemented with 1 mM IPTG and 1 μm ferrioxamine B **(B)** or with 50 mM FeCl$_3$ **(C)**. Proteins were separated by SDS-PAGE and immunoblotted against FoxR using the FoxR$^{peri}$ polyclonal antibody (upper panel) and FoxA using the FoxA$^{SD}$ polyclonal antibody (middle panel). Detection of the outer membrane lipoprotein OprL was used as loading control. Positions of the protein fragments and the molecular size marker (in kDa) are indicated. Presence of the HA-tag adds ∼1 kDa to the molar mass of the protein fragments. Blots are representatives of 3 biological replicates ($N = 3$). The raw data underlying the graphs shown in the figure can be found at Mendeley Data repository (Mendeley Data, V1, 10.17632/nxh4c8ymnn.2). Western blot can be found in S1 Raw Images.
(TIF)

**S3 Fig. Detection of the *P. aeruginosa* FoxR protein by western blot and complementation of the *P. aeruginosa* Δ*foxR* mutant. (A)** *P. aeruginosa* PAO1 wild-type strain bearing the pMMB67EH empty plasmid or its derivative expressing the FoxR protein were grown under iron-restricted conditions and 1 mM IPTG. Proteins were immunoblotted against the anti-$\sigma$ factor FoxR using a polyclonal antibody. Positions of the protein fragments and the molecular size marker (in kDa) are indicated. **(B)** The *P. aeruginosa* PAO1 wild-type strain and its isogenic Δ*foxR* mutant bearing the *foxA*::*lacZ* transcriptional fusion and the pMMB67EH empty (-) or the pMMB67EH-derived plasmid producing the FoxR, FoxR-HA, or HA-FoxR proteins (S1 Table) were grown under iron-limitation conditions and in absence (-) or presence (+) of 1 μm ferrioxamine B. Activity was determined by β-galactosidase assay and data are means ± SD from 3 biological replicates ($N = 3$). *P*-values were calculated by two-tailed *t* test by comparing the value obtained in the PAO1 wild-type strain with that of the mutant strains in the same growth condition and are represented in the graphs by *, $P < 0.05$; **, $P < 0.01$; ***, $P < 0.001$; and ****, $P < 0.0001$. The raw data underlying the graphs shown in the figure can be found at Mendeley Data repository (Mendeley Data, V1, 10.17632/nxh4c8ymnn.2). Western blot can be found in S1 Raw Images.
(TIF)

**S4 Fig. Analysis of FoxR and FoxA$^{SD}$ protein stability.** *E. coli* BL21 cells producing the indicated protein domain were grown as described in Materials and methods. Cultures were harvested, resuspended in buffer A at different pHs, and subjected to sonication. The gel shows protein samples from the soluble (S) and the pellet (P) fractions after cells were resuspended in buffer A at pH 8.0, while other trials with various pHs showed similar results. Position the

molecular size marker (in kDa) is indicated. Western blot can be found in S1 Raw Images.
(TIF)

**S5 Fig. Structural model of FoxR. (A)** The Alphafold protein structure of FoxR (AF_AF-Q9I115F1) is shown. The different domains of the protein are indicated. **(B, C)** Structural alignment between the periplasmic portions of FoxR and PupR (6OVM) with focus on the 2 anti-parallel β strands of the STN domain **(C)**.
(TIF)

**S6 Fig. Structural homology between FoxR$^{STND}$ and FoxA$^{SD}$.** Structural features of the FoxR$^{STND}$ **(A)** and FoxA$^{SD}$ **(B)** protein structures (obtained from AF_AFQ9I115F1 and 6I97, respectively). **(C)** Structural alignment between both protein domains.
(TIF)

**S7 Fig. Structural prediction of the FoxR$^{peri}$/FoxA$^{SD}$ complex. (A)** Structural alignment between the Alphafold FoxR$^{peri}$/FoxA$^{SD}$ predicted complex (in blue and orange) and the PupB/PupR complex (in green, 6OVK) with an RSMD of 0.862. **(B)** Alphafold model for the FoxR$^{peri}$/FoxA$^{SD}$ complex colored by its pLDDT value. **(C)** PAE chart for the FoxR$^{peri}$/FoxA$^{SD}$ model showing a likely interaction between FoxA$^{SD}$ and the FoxR$^{peri}$ proteins. The raw data underlying the graphs shown in the figure can be found at Mendeley Data repository (Mendeley Data, V1, 10.17632/nxh4c8ymnn.2).
(TIF)

**S1 Table. Bacterial strains and plasmids used in this study.**
(PDF)

**S2 Table. Sequences of the primers used in this study.**
(PDF)

**S1 Raw Images. Original, uncropped, and minimally adjusted western blot images from this study.**
(PDF)

## Acknowledgments

We thank T. Krell for assistance with protein purification, and C. Civantos and A. Sánchez-Jiménez for technical assistance with western blot experiments.

## Author Contributions

**Conceptualization:** María A. Llamas.

**Data curation:** Sarah Wettstadt, Francisco J. Marcos-Torres, Álvaro Ortega, María A. Llamas.

**Formal analysis:** Sarah Wettstadt, Francisco J. Marcos-Torres, Álvaro Ortega, María A. Llamas.

**Funding acquisition:** María A. Llamas.

**Investigation:** Sarah Wettstadt, Francisco J. Marcos-Torres, Joaquín R. Otero-Asman, Álvaro Ortega, María A. Llamas.

**Methodology:** Sarah Wettstadt, Francisco J. Marcos-Torres, Joaquín R. Otero-Asman, Alicia García-Puente, Álvaro Ortega.

**Project administration:** María A. Llamas.

**Supervision:** María A. Llamas.

**Validation:** Sarah Wettstadt, Francisco J. Marcos-Torres, Álvaro Ortega, María A. Llamas.

**Writing – original draft:** Sarah Wettstadt, Francisco J. Marcos-Torres, María A. Llamas.

**Writing – review & editing:** Francisco J. Marcos-Torres, Álvaro Ortega, María A. Llamas.

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
