## [Editor Report · Decision Letter 0]

4 Aug 2024

Dear Dr Llamas, 

Thank you for submitting your manuscript entitled "New insights into the bacterial cell-surface signalling mechanism: the signalling domain of the receptor is cleaved and protects the anti-σ factor from proteolysis" for consideration as a Research Article by PLOS Biology.

Your manuscript has now been evaluated by the PLOS Biology editorial staff, as well as by an academic editor with relevant expertise, and I am writing to let you know that we would like to send your submission out for external peer review. However, in the current form, without showing the exact activation mechanism, we can only invite for submission as a Short Report. 

However, before we can send your manuscript to reviewers, we need you to complete your submission by providing the metadata that is required for full assessment. To this end, please login to Editorial Manager where you will find the paper in the 'Submissions Needing Revisions' folder on your homepage. Please click 'Revise Submission' from the Action Links and complete all additional questions in the submission questionnaire. Please, when adding the rest of the metadata choose "Short Report".

Once your full submission is complete, your paper will undergo a series of checks in preparation for peer review. After your manuscript has passed the checks it will be sent out for review. To provide the metadata for your submission, please Login to Editorial Manager (https://www.editorialmanager.com/pbiology) within two working days, i.e. by Aug 06 2024 11:59PM.

Kind regards,

Melissa

Melissa Vazquez Hernandez, Ph.D.

Associate Editor

PLOS Biology

---

## [Decision Letter · Decision Letter 1]

12 Sep 2024

Dear Maria

Thank you for your patience while your manuscript "New insights into the bacterial cell-surface signalling mechanism: the signalling domain of the receptor is cleaved and protects the anti-σ factor from proteolysis" went through peer-review at PLOS Biology. Your manuscript has now been evaluated by the PLOS Biology editors, an Academic Editor with relevant expertise, and by two independent reviewers.

As you will see in the reports, the reviewers have some suggestions and concerns that should be addressed. Both reviewers have concerns regarding the AUC data. Furthermore, as noted by Reviewer #2 the discussion of the structure prediction should emphasize that the model is a prediction, without experimental validation. All reviewers’ suggestions regarding modifications on the text and the figures should be addressed. Additionally, after discussion with the Academic Editor we can only accept the study as a Short Report without additional experimental data, as we have discussed previously, so please adjust the number of main figures to 4. 

**IMPORTANT - SUBMITTING YOUR REVISION**

*Resubmission Checklist*

*Published Peer Review*

*PLOS Data Policy*

*Blot and Gel Data Policy*

Sincerely,

Melissa

Melissa Vazquez Hernandez, Ph.D.

Associate Editor

PLOS Biology

REVIEWERS' COMMENTS:

Reviewer #1: 

The acquisition of iron by the uptake of siderophores in Gram-negative bacteria implies a Cell-Surface Signaling (CSS) process that transfers the extracellular environmental signal (the presence of the iron-siderophore) to the cytoplasm via a number of proteins localized in the outer membrane, the periplasm and the inner membrane, finally resulting in the release of a sigma factor that, in the absence of the signal, remains trapped by a membrane-bound anti-sigma factor. This sigma factor drives the RNA polymerase to a number of specific genes. The molecular details of ferric siderophore-mediated activation of the iron import machinery by the CSS process are complex and not fully clear. Using biochemical and gene expression (two-hybrids) approaches, and supported by structural models, in this manuscript authors provide experimental evidence that allows proposing some changes in the view on how the signal transducing process occurs. Previously, the model implied that the presence of the signal allows the interaction between the siderophore receptor located in the outer membrane (TBDT) and the anti-σ factor in the periplasm, promoting the proteolysis of the anti-sigma factor and in turn releasing the sigma factor. The results presented here allow proposing a somewhat different way of signal transduction in which sigma factor release does not depend on this interaction. Rather, the contact between the TBDT and the anti-σ factor would already occur in the absence of the signal (the iron-siderophore). This interaction would protect the anti-sigma from proteolysis by periplasmic proteases. To exert this role, the anti-sigma undergoes a self-catalysed proteolytic cleavage. Signal transduction involves further cleavage of the anti-sigma, probably by the CtpA protease. Initial evidence is provided for three different siderophore receptors in two different Pseudomonas species, and the molecular mechanisms are later worked out in more detail for one of them (the P. aeruginosa ferrioxamine FoxA receptor/transducer and the FoxR anti-sigma), defining the interacting domains.

In my view, the experiments and the results are clear, rigorous and convincing, and provide useful information. The manuscript is clearly written and is easy to follow in spite of the complexity of the problem.

Specific comments

1. Figure 1 was missing in the "R1" version of the manuscript that I received. Fortunately, the original "R0" version was also available to me at the PLoS web, and shows this figure, which is essential to follow the manuscript.

2. Page 7, line 144 onwards. Authors overproduce the signalling domain of the siderophore receptor and detect a resulting phenotype (inhibition of the CSS activity). I assume that this overproduction is done in a way that allows the signalling domain to be exported to the periplasm, but this is not specifically explained in the text. While this is probably obvious for the authors, it is not so for the readers, and it is important to know it to follow the line of reasoning. Please explain.

3. Page 11, lines 241 and 249. Where authors include a call to Fig. 5B, they probably mean Fig. 5A. In addition, in line 258, the call to Fig. 5C seems to correspond to Fig. 5B (there is no panel C) 

4. I am not familiar with sedimentation velocity analytical ultracentrifugation, so that my comment might be wrong, but please note that the sedimentation coefficient values indicated in the text (page 11) for the different proteins analysed are always twice higher than those that I would (perhaps incorrectly) deduce from Fig. 5A. Please check the scale of the X-axis, or disregard this comment.

Reviewer #2: 

Summary:

Gram-negative bacteria utilize TonB-dependent outer-membrane receptors (TBDRs) to acquire exogenous nutrients from their environments. Among such cargo is iron, often in the form of iron chelates sequestered within siderophores. Many of these TBDRs play accessory roles in signal transduction, upregulating their own production in the presence of their substrate. These TonB-dependent transducers (TBDTs) contain an additional N-terminal signaling domain (SD) which participates in the pathway, ultimately resulting in release of a sigma factor which aids upregulation of transcription for the TBDT. While certain aspects of this pathway are understood, questions remain about the interactions between the TBDT and its cognate anti-sigma factor, as well as the specific role of the SD within the pathway.

The current manuscript by Wettstadt et al builds on the current signaling model by demonstrating that in the absence of the inducing signal, the SD of one TBDT, FoxA, interacts with the cognate anti-sigma factor, FoxR, stabilizing it and preventing proteolytic degradation and subsequent sigma factor release. Using a two-hybrid binding assay, they identified specific interacting domains in the proteins. The authors used AlphaFold to predict the structure of the FoxA-SD/FoxR complex, and from this, identified candidate amino acid residues in the proteins which are important to the binding interface. The same two-hybrid system was used to validate key residues. Lastly, they show that the SD of FoxA is cleaved in a process involving CtpA and Prc, building off previous work from this group which described the role of Prc and CtpA in the signaling process. 

The findings presented here present an important step in understanding signal transduction by TBDTs. However, the manuscript in its current form is not suitable for publication in PLOS Biology and the impact of this work on the field is not necessarily well suited to PLoS Biology. Several issues are outlined below, which the authors are invited to consider prior to resubmission.

1. Figure 6 and its associated results section cannot stand alone and be discussed as though it is experimental data. The model shown in this figure was generated using AlphaFold and is therefore a predicted model only, so the results from the prediction must be discussed and considered in this context. The information presented in this section can stay, but a suggestion is to revise the figure and use it to show a comparison between the current AlphaFold model and the already solved crystal structure of the PupB/PupR complex. In this depiction, the authors can highlight similarities between the solved PupB/PupR interface and the predicted FoxA/FoxR interface. This then naturally flows into the data from figure 7 in which important residues are directly tested and validated. Regardless of how the authors choose to address this, if the AlphaFold model is to remain in the manuscript, the supplementary information should include both a portrayal of the model colored according to B-factor, highlighting the relevant domain(s), as well as any relevant PAE matrices. The text sections as well should be modified to reflect that these are inferences from a predicted model, not "structural modeling data".

2. Concerning the reworked figure 6, please modify the color scheme and label font/placement. The current version is difficult to interpret in its current from.

3. The results section concerning the AUC data presented in figure 5 is not interpretable in its current form. My combined PDF shows panels A and B for this figure, but the text does not refer to 5A at any point and refers to panel 5C which does not exist. It is not clear if this is simply a continuity/labeling error in the text because this section also refers to protein species with sedimentation coefficients which do not correspond to what is shown in the figure. For example, line 241 refers to FoxR-peri at 2.1 S, but this does not seem consistent with what is shown in the figures. This section should be revised for consistency and clarity.

4. For figure 5A, change the labels to reflect the actual protein domains used. FoxR-peri and FoxA-SD, not simply FoxR and FoxA.

5. Figure 4C is out of place. The results section describing these data is a full 4 pages after the section describing 4A and B. Further, interpretation of 4C requires the reader to have already seen figure 6 in order to know why the STND domain and the relevant stretch of amino acids is important. This panel should be removed from figure 4, and in my opinion added to figure 7 instead.

6. Throughout the results section, be mindful of proper use of past tense to describe the results of the experiments. The blot showed that something happened, the assay demonstrated, etc. However, statements about interpreting the results ("these results suggest…") can and should be in present tense.

7. For figure 7A, move the "WT like" FoxR-C / FoxA-SD data point to either the top of the graph or the bottom with the other controls.

8. Line 368 - I'm not sure I agree with the claim that a single protein band is seen in the PAO1 sample + ferrioxamine. It is faint, but the same lower MW band as is seen in the Δprc + ferrioxamine lane is visible. The overall conclusion that absence of Prc reduces CSS activity, presumably due to loss of the SD from FoxA as suggested by the western blot, remains valid.

9. Loading controls should be shown for the blots in Figure 2B and 8

---

## [Editor Report · Decision Letter 2]

22 Oct 2024

Dear Dr Llamas,

Thank you for your patience while we considered your revised manuscript entitled "New insights into the bacterial cell-surface signalling mechanism: the signalling domain of the receptor is cleaved and protects the anti-σ factor from proteolysis" for publication as a Short Reports at PLOS Biology. This revised version of your manuscript has been evaluated by the PLOS Biology editors and by the Academic Editor.

Based on our Academic Editor's assessment of your revision, we are likely to accept this manuscript for publication, provided you satisfactorily address the data and other policy-related requests stated below. In addition, the Academic Editor thinks that Reviewer 2's point 1 should be addressed better. The Academic Editor believes that the coordinates from experimental data are truly "structural models" that fit the experimental data, thus you should identify the AlphaFold structures as "structure predictions", "predicted structures", "predicted models", "AlphaFold models", "AlphaFold predictions" or other such terminology rather than calling them structural models.

We would also like you to consider a suggestion to improve the title, but please let us know if it is not accurate:

"Bacterial TonB-dependent receptors interact with anti-σ factor in absence of environmental signal to protect it from proteolysis"

We expect to receive your revised manuscript within two weeks. 

*Published Peer Review History*

*Press*

Sincerely,

Ines

--

Ines Alvarez-Garcia, PhD

Senior Editor

PLOS Biology

on behalf of

Melissa Vazquez Hernandez, Ph.D.

Associate Editor

PLOS Biology

Fig. 1A; Fig. 2B, C; Fig. 3B, C; Fig. 4A; Fig. S3B and Fig. S7C

Please also ensure that figure legends in your manuscript include information ON WHERE THE UNDERLYING DATA CAN BE FOUND, and ensure your supplemental data file/s has a legend.

***Please also make the data you have deposited in Mendeley (https://data.mendeley.com/preview/nxh4c8ymnn?a=001b68a8-d935-4e26-ad30-397022651f62) publicly available at this stage. If the raw data requested is included there, please note in all the corresponding figure legends that "The data underlying the graphs shown in the figure can be found at https://data.mendeley.com/preview/nxh4c8ymnn?a=001b68a8-d935-4e26-ad30-397022651f62 "

CODE POLICY

We require the original, uncropped and minimally adjusted images supporting all blot and gel results reported in an article's figures or Supporting Information files. We will require these files before a manuscript can be accepted so please prepare and upload them now. Please carefully read our guidelines for how to prepare and upload this data: https://journals.plos.org/plosbiology/s/figures#loc-blot-and-gel-reporting-requirements

You should include the raw gels shown in the following figures: Fig. 1B; Fig. 4B, C; Fig. S2B, C; Fig. S3A and Fig. S4

---

## [Editor Report · Decision Letter 3]

31 Oct 2024

Dear Dr Llamas,

Thank you for the submission of your revised Short Reports "Bacterial TonB-dependent transducers interact with the anti-σ factor in absence of the inducing signal protecting it from proteolysis" for publication in PLOS Biology. On behalf of my colleagues and the Academic Editor, Ann Stock, I am pleased to say that we can in principle accept your manuscript for publication, provided you address any remaining formatting and reporting issues. These will be detailed in an email you should receive within 2-3 business days from our colleagues in the journal operations team; no action is required from you until then. Please note that we will not be able to formally accept your manuscript and schedule it for publication until you have completed any requested changes.

IMPORTANT: Thank you for attending the previous requests. However some of the requests did not meet fully the requirements. We are missing the raw data for Figs Fig 2C, S3B and S7C, as well as the raw gels for Figs 4B, S3A and S4. Additionally, as mentioned before, please also ensure that figure legends in your manuscript include information ON WHERE THE UNDERLYING DATA CAN BE FOUND; e.g. "The data underlying the graphs shown in the figure can be found at Mendeley Data repository (Mendeley Data, V1, doi: 10.17632/nxh4c8ymnn.1).". This should be in every figure that required raw data.

PRESS

Sincerely, 

Melissa

Melissa Vazquez Hernandez, Ph.D., Ph.D.

Associate Editor

PLOS Biology
